# An Algorithm for Individual Dosage in Cadmium–Zinc–Telluride SPECT-Gated Radionuclide Angiography

**DOI:** 10.3390/diagnostics11122268

**Published:** 2021-12-04

**Authors:** Maria Normand Hansen, Christian Haarmark, Bent Kristensen, Bo Zerahn

**Affiliations:** 1Department of Nuclear Medicine, Copenhagen University Hospital Herlev and Gentofte, 2730 Herlev, Denmark; christian.eickhoff.haarmark.nielsen.01@regionh.dk (C.H.); bent.kristensen.01@regionh.dk (B.K.); Bo.Zerahn@regionh.dk (B.Z.); 2Department of Clinical Medicine, University of Copenhagen, 2200 Copenhagen, Denmark

**Keywords:** personalised medicine, RNA, CZT SPECT, gated SPECT

## Abstract

The aim of the present study was to test an individualised dose without compromising the ease of analysing data when performing equilibrium radionuclide angiography (ERNA) using cadmium–zinc–telluride (CZT) SPECT. From March 2018 to January 2019, 1650 patients referred for ERNA received either an individualised dose of ^99m^Tc-labeled human serum albumin (HSA) according to their age, sex, height, and weight (*n* = 1567), or a standard dose of 550 MBq (*n* = 83). The target count rate (CR_T_) was reduced every two months from 2.7 to 1.0 kcps. A final test with a CR_T_ of 1.7 kcps was run for three months to test whether an agreement within 2% points for the determination of LVEF, on the basis of only two analyses, was obtainable in at least 95% of acquisitions. All the included ERNAs were performed on a dedicated cardiac CZT SPECT camera. When using the algorithm for an individualised dose, we found that agreement between the measured and predicted count rate was 80%. With a CR_T_ of 1.7 kcps, the need for more than two analyses to obtain sufficient agreement for LVEF was 4.9%. Furthermore, this resulted in a mean dose reduction from 550 to 258 MBq. Patients’ weight, height, sex, and age can, therefore, be used for individualising a tracer dose while reducing the mean dose.

## 1. Introduction

Technological advances in cancer treatment and screening procedures have resulted in an increase in long-term cancer survivors [1,2,3]. This has led to an increased need for managing the potential long-term side effects of chemotherapy and radiotherapy, among which, cardiovascular complications are frequent [2,3]. Therefore, accurate monitoring of cardiac function during potentially cardiotoxic chemotherapy is of importance. Patients undergoing potentially cardiotoxic chemotherapy are often monitored by quantifying left ventricular ejection fraction (LVEF). Several consecutive measurements are used when monitoring for cardiotoxic effects and for subsequent therapeutic decisions.

Chemotherapy-induced cardiac damage may be caused by a direct toxic effect or as an accelerated development of cardiovascular disease and appears up to years after the initiation of treatment [4,5].

Several techniques are currently available for this monitoring of LVEF [3,6]. Equilibrium radionuclide angiography (ERNA) is a well-validated non-invasive test [7], and in particular, cadmium–zinc–telluride single-photon emission tomography (CZT SPECT) radionuclide angiography ranks high, due to practicability, operator independency, and reproducibility [8,9,10].

Since radionuclide angiography was implemented for clinical use in the 1980s, development has moved toward shorter acquisition times and lower doses of tracer. In 2016, Duvall et al. reported that an injected dose of tracer may be reduced by up to 50% without compromising the image quality [9]. The new dedicated cardiac cameras with CZT detectors and individualising the activity administered to each patient should, therefore, enable a reduction in radiation exposure even further [8,9,10,11,12,13,14,15].

Previous studies have already evaluated the possibility of using personalised models when planning doses for a radionuclide-based assessment of cardiac function. These studies have primarily focused on myocardial perfusion imaging [11,12,13,16,17], but one study has evaluated planar ERNA where the dose was based upon the patient’s physical variables according to a chart and not a formula, which was declared to be a weakness of the method [18].

When performing CZT SPECT-gated radionuclide angiography, it has previously been shown that the patients’ height, weight, sex, and age may explain up to 75% of the variation in the count rate [19,20]. The next step, as suggested in an editorial by Joris D. van Dijk [21], is to test this information in a clinical setting. Subsequently, the aim of this study was to test dose planning in an everyday setting [19,20] with the goal of reducing radiation exposure without compromising the ease of processing data.

## 2. Materials and Methods

### 2.1. Population

From March 2018 to January 2019, a total of 1696 patients were referred for routine assessments of LVEF. Patients received either an individualised dose according to their weight, height, sex, and age, or a standard dosage of 550 MBq if data on weight and height were insufficient upon referral. Patient flow is illustrated in Figure 1, and detailed information on target count rate groups is shown in Table 1.

Sex, age, blood pressure, heart rate, body weight, and body height were registered, and information on anthropometric data, cardiac variables, dosage information, and frequency of need for more than two analyses is provided in Table 2. The study was conducted according to the guidelines of the Declaration of Helsinki and approved by the Institutional Review Board (Herlev Gentofte Hospital Directional Board, 26 February 2021, WorkZone number: 20076870).

Anonymised data used in the current study are available from the corresponding author.

### 2.2. Image Acquisition and Processing

All the acquisitions were performed at the Department of Nuclear Medicine at Herlev-Gentofte Hospital. ERNAs were performed on a dedicated cardiac CZT SPECT gamma camera, GE Discovery 530c (GE Healthcare, Milwaukee, WI, USA). Each subject was either given an individualised dose (see below) of ^99m^Tc-labeled human serum albumin (HSA) intravenously on the basis of the algorithm below, or 550 MBq if data on patient height and body weight were unavailable, as noted above. An acquisition protocol for multigated acquisition, using 16 bins per R-R interval, requesting 600 accepted beats, and a 20% energy window centred on 140 keV, was carried out.

Count rate was read from the work screen during acquisition and calculated as the mean of three readings (one at the beginning, middle, and end).

For image analyses, we used a Xeleris 3 Imaging workstation reorientation software (GE Healthcare, Milwaukee, WI, USA, version no. 3.0562) and Cedars-Sinai QBS processing software (Cedars-Sinai, Los Angeles, CA, USA, revision 2009.0).

Each acquisition was analysed by two experienced technologists independently of one another, and mean values of the cardiac variables were calculated. If the calculated LVEF varied by more than 2 percentage points, further analyses were performed until a sufficient agreement was obtained. Depending on the workflow circumstances, subsequent analyses were performed either by one of the two first technologist, a third technologist, or if necessary a physician. Apart from LVEF, data on left ventricular end diastolic and systolic volumes (EDV and ESV, respectively) were also recorded. Tracer dose was registered, and an adjustment was performed for differences between the scheduled and actual time of injection to the nearest 5 min.

### 2.3. Optimum Target Count Rate Identification and Equation Selection

The following two equations for calculating patient tailored dose of ^99m^Tc-labeled HSA were developed (*D_P_* = planned dose, *X_W_* = weight, *X_H_* = height, *X_G_* = sex, *X_A_* = age, *CR_T_* = target count rate):

Equation (1) [20]:(1)$$D_{P}=550MBq-\left(EXP\left(1.8-0.028X_{W}+0.00005X_{W}{}^{2}+0.0087X_{H}+0.06X_{G\;}-0.0021X_{A}\right)-CR_{T}\right)/0.00756$$

Equation (2) [19]:(2)$$D_{P}=550MBq\times CR_{T}/EXP\left(1.621-0.020X_{W}+0.008X_{H}+0.06X_{G}-0.002X_{A}\right)$$

Both equations were derived on the basis of previous examinations of a large patient group who all received a fixed dose of tracer (550 MBq ^99m^Tc-labeled HSA) [19,20]. The main difference between the two equations is the insertion of an extra element in equation 1 with body weight squared.

Equation (1) was tested on the first group of patients, aiming at a target count rate of 2.7 kcps for a period of 2 months running through March and April 2018. However, this equation alone often suggests negative doses for patients with the lowest body weights, as illustrated in Figure 2, where the suggested patient doses for each equation is plotted against patient weight. In such cases, a minimum dose of 150 MBq was used (see below). On the other hand, Equation (2) suggests high doses in patients with a high body weight (Figure 2). In the large group of patients with a more average body weight, there were only minor differences between the two equations with regard to the suggested dose. As a consequence, the two equations were combined in to one algorithm that was used from May 2018 onwards, using Equation (1) in patients with a body weight ≥ 73 kg and Equation (2) for those with a body weight < 73 kg.

Different target count rates were each tested for two months at a time, moving gradually downwards from 2.7 to 1.0 kcps (see Figure 1 and Table 2). This was performed in order to identify a level for the target count rate, where the frequency of two analyses of the same acquisition reaching an agreement of LVEF within 2 percent points was approximately 95%. Hereafter, target count rate was set to 1.7 kcps to test if this was appropriate to meet the requested frequency of sufficient agreement between analyses.

Patients were injected with a minimum dose of 150 MBq when aiming at count rates of 2.7 (19 (7.6%)) and 1.5 kcps (15 (5.2%)), and 100 MBq when aiming at a count rate of 1.0 kcps (36 (11.5%)) (Table 1). This was in order to stay above the minimum activity recommended for children with a 20% safety margin according to EANM dose recommendations [7]—a necessary precaution to avoid low, and also occasionally negative, doses suggested when using Equation (1) alone (target count rate 2.7) and in order to avoid the need for frequent re-injections when aiming at count rates of 1.5 and 1.0 kcps. Accordingly, the need to use the minimum dose was 0 (0%) and 1 (0.004%) for target count rate groups 1.7 and 2.0, respectively.

### 2.4. Subgroup Analyses

In 122 of the assessments (32 with target count rate of 1.0 kcps and 90 with target count rate 1.7 kcps), the remaining activity in needles and syringes was measured after injection of tracer in order to evaluate how much of the scheduled tracer activity was left in these utensils, in order to evaluate the subsequent potential influence on count rate.

### 2.5. Statistical Analyses

All statistical analyses were conducted with IBM Corp. Released 2017. IBM SPSS Statistics for Windows, Version 25.0. Armonk, NY, USA: IBM Corp.

Scatter plots were made in RStudio, RStudio Team (2019). RStudio: Integrated Development for R. RStudio, Inc., Boston, MA, USA, available on http://www.rstudio.com/products/rstudio/download/#download (accessed on 9 September 2020).

Group comparison was performed with ANOVA and Bonferroni correction for multiple comparisons and a *t*-test for comparisons between two groups.

Linear regression analysis (Pearson correlation coefficient) was used to quantify the degree of association between measured and predicted count rate in all included patients.

Level of significance was set at a 5% level.

## 3. Results

### 3.1. Population

A total of 1567 patients received individualised activity protocol, and 83 received a standard activity of 550 MBq due to insufficient data on their height and weight upon referral. In 27 cases, acquisitions were excluded because NaI detector SPECT or planar was performed instead, 18 cases were excluded if the patient received another dose than scheduled, and 1 case was excluded due to an injection of 18-F-FDG prior to ERNA.

The patients who received an individualised dose or standard dose (*n* = 1650) for ERNA received treatment for breast cancer, 697 (42.2%); leukaemia, 24 (1.4%); lymphoma, 245 (14.9%); sarcoma, 225 (13.6%); renal cancer, 108 (6.6%); hepatocellular carcinoma, 47 (2.9%); ovarian cancer, 53 (3.2%); malignant melanoma, 174 (10.6%); and other cancers or more than one type of cancer, 77 (4.7%).

There were no significant differences with regard to sex frequency, age, height, weight, heart rate at rest, and blood pressure between the patients in the five different target group periods (Table 2, rows 1 to 6).

Although the analysis of variance (ANOVA) between the target group periods indicated significant differences for end diastolic volume (EDV) and LVEF (*p* = 0.028 * for EDV, *p* = 0.053 for end systolic volume (ESV), *p* = 0.017 * for LVEF), there were no significant differences between group analyses after the Bonferroni correction, indicating the risk of a possible error due to multiple comparisons. Despite this, the apparently increasing volumes for both EDV and ESV with the increasing mean dose could indicate some degree of partial volume effect. Nevertheless, there is no similar indication of changes in LVEF with the changes in the target count rate (Table 2, rows 7 to 9).

### 3.2. Test of Patient Tailored Algorithm

When plotting all the measured count rates against the predicted count rates (*n* = 1650), we found an excellent prediction of the count rate, as illustrated in Figure 3. Linear regression showed a slope coefficient of 1.012 and an intercept of 0.007 kcps. The algorithm based on information on patient weight, height, sex, and age explained 80% (R^2^ = 0.80) of the variation in the observed count rate, which is an improvement from the expected 75%.

When using the combination of Equations (1) and (2) as described, we were able to obtain a mean predicted count rate very close to the measured count rate (0 to 4.4%) throughout the range of target count rates from 1.0 to 2.7 kcps while reducing the mean dose as the target count rate was lowered (Table 2, rows 10 to 13).

The gradual reduction in the target count rate led to a frequency of disagreement of more than 2 percent points between the LVEF analyses of an ERNA above 5% when aiming at count rates of 1.5 and 1.0 kcps. This led to the decision to run the final test period from November 2018 to the end of January 2019, aiming at a count rate of 1.7 kcps. With this target count rate, a sufficiently low frequency of disagreement between the two first analyses of LVEF was obtained (21 out of 432 or 4.9%) (Table 2, row 14).

An individualised dose reduced the overall mean dose by 53% (from 550 to 258 MBq) when aiming at a count rate of 1.7 kcps, as compared to our standard dose of 550 MBq (Table 2, rows 10 and 11).

The mean absolute (numerical) time between the scheduled and actual tracer injection was 9 min, ranging from 0 to 140. This gave rise to a mean absolute difference between the scheduled dose and the actual dose at injection of 5 MBq (2%), ranging from 0 to 114 MBq, or up to 29%.

### 3.3. Subgroup Analyses

In the injection utensils used for target count rates of 1.0 and 1.7 kcps, there was no difference in the remaining activity (9.6 ± 4.4 vs. 10.4 ± 5.4 MBq, respectively, *p* = 0.43 for *t*-test with unequal variance). However, the relative remaining activity was higher in the utensils used for a target count rate of 1.0 than those used for a target count rate of 1.7 kcps (5.5 ± 3.1% vs. 4.3 ± 2.7%, respectively, *p* = 0.0083 for *t*-test with unequal variance). The maximum remaining amount of tracer in the injection utensils was 16.5%.

## 4. Discussion

This study successfully tested a combination of two equations for individualised dose planning in CZT ERNA, allowing for a reduction in the mean dose of at least 50% (to a mean effective dose of 1.6 mSv (range: 0.6 to 3.6 mSv)), without compromising the ability to produce reproducible analyses of LVEF. The average dose using a target count rate of 1.7 kcps is approximately half of the current recommendation for planar nuclear angiography [7,22], and as suggested possible by Duvall et al. [9].

Additionally, we found a level of the target count rate that minimises the need for a lower dose limit. When doing this, we used the frequency of disagreement of more than 2 percent points between the two LVEF calculations as a surrogate for image quality. Nonetheless, this limit is used in daily clinical practice and is subsequently practical in this context. While using the above-mentioned algorithms in daily practice, there were two instances in young males where the suggested dose was below the lower limit of 100 MBq. In response, we altered the weight limit used to decide which of the two equations to use from 73 to 82 kg (effectuated April 2020). This has been the only alteration that we have found it necessary to perform.

There are several benefits to reducing the administered dose of radiopharmaceuticals. Primarily, optimising the exposure to radiation according to the “as low as reasonably achievable (ALARA) principles” would reduce the radiation burden to both patients and staff in accordance with both guidelines [23] and national legislation. Additionally, a more uniform count rate should improve the uniformity of test results, regardless of the patients’ size, by eliminating the consequences of a partial volume effect caused by higher count rates in smaller patients versus lower count rates in patients with larger body proportions as well as the possibility to adapt the dose to the individual patient despite changes in body weight during therapy [24]. In the present study, however, this partial volume effect appears to be of minor importance since it has implications on EDV and ESV but not LVEF. As such, it is also unlikely that individualised tracer dosage has influenced the accuracy of the CZT SPECT ERNA procedure.

Administering an overall lower dose to each patient will inevitably reduce the costs of both the tracer and ^99m^Tc without altering the procedure, particularly with regard to scan time, although this doubles the need for more than two analyses to approximately 5%. As an example, the number of vials per 100 performed ERNAs was, in our department, 39.8 in 2016, when the standard dose of 550 MBq per acquisition was used, as compared to 27.3 in 2019, when a patient tailored dose aiming at a target count rate of 1.7 kcps was used routinely, with an average cost reduction of 31.5%. It is, however, mandatory that clinicians upon referral for ERNA provide sufficient information on weight and height for the computed calculation of an individual tracer dose. This has required some adaptation for the clinicians, which is reflected in Table 1, where the need for the use of the standard dose of 550 MBq decreased from 41 and 30 in target count rate groups 2.7 and 2.0 kcps, respectively, to below 10 in the later series.

The individualised planning of doses and the on-the-fly monitoring of count rates makes it possible to pinpoint cases of a paravenous injection as the examination is running due to a lower count rate than expected. In this study, this happened in 10 out of 1650 cases.

There is an apparent improvement in the predictability of the count rate from the expected 75% as previously published to 80%. This may well be attributed to the use of a combination of the two dose calculation equations instead of only using one, thus avoiding their respective weaknesses at each end of the spectrum of body weight (Figure 2). Another factor that may have marginally contributed is the adjustment for the exact tracer injection time instead of just using the scheduled time. Still, the mean deviation from the scheduled dose due to differences between the actual and planned injection time was only 4.3 MBq (1.6%).

Variations due to unknown residual activity in injection utensils; differences in thorax shape; and the effect of varying tissue amounts around the heart due to prior surgery—left sided mastectomy, in particular—may account for other unexpected major deviations from the expected count rate [25]. It can be expected that the relative residue of tracer in injection utensils will increase with lower target doses as long as the same dilution of tracer is used.

## 5. Conclusions

Patients’ weights, heights, sexes, and ages can be used to individualise the planning of their tracer doses using a combined algorithm with a target count rate. The benefits of this approach are numerous—most importantly, the radiation burden to both patients and staff can be reduced by at least 50% compared to that for the hitherto-used standard dose of 550 MBq and, subsequently, even further than the current recommendations for planar nuclear angiography in accordance with ALARA principles. Secondary benefits include reduced costs in terms of the tracer and technetium; encountering shortages of technetium; the possibility of identifying a paravenous injection on the fly; and, potentially, an increased uniformity of test results across patient age, sex, height, and weight as well as changes, particularly in the latter.

## Figures and Tables

**Figure 1 diagnostics-11-02268-f001:**
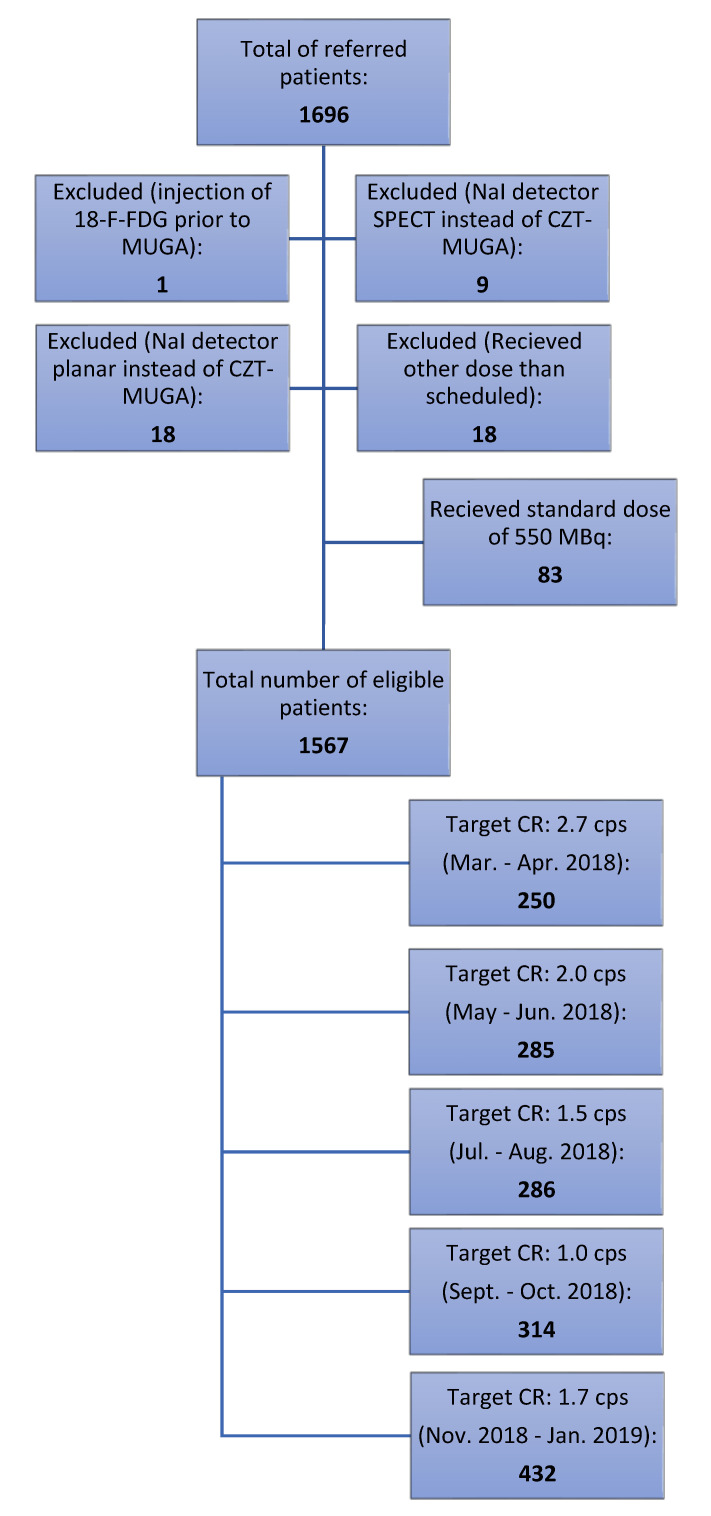
Flow chart of patients in and excluded from the study.

**Figure 2 diagnostics-11-02268-f002:**
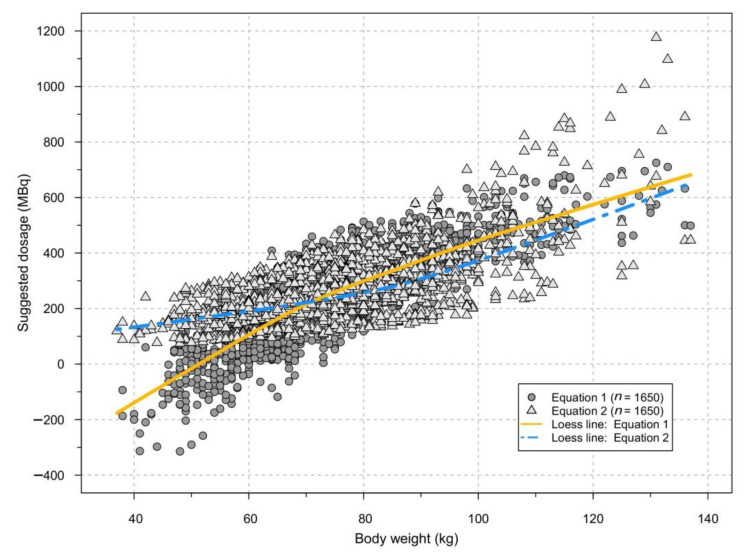
Suggested dosage plotted against weight, depending on equation used.

**Figure 3 diagnostics-11-02268-f003:**
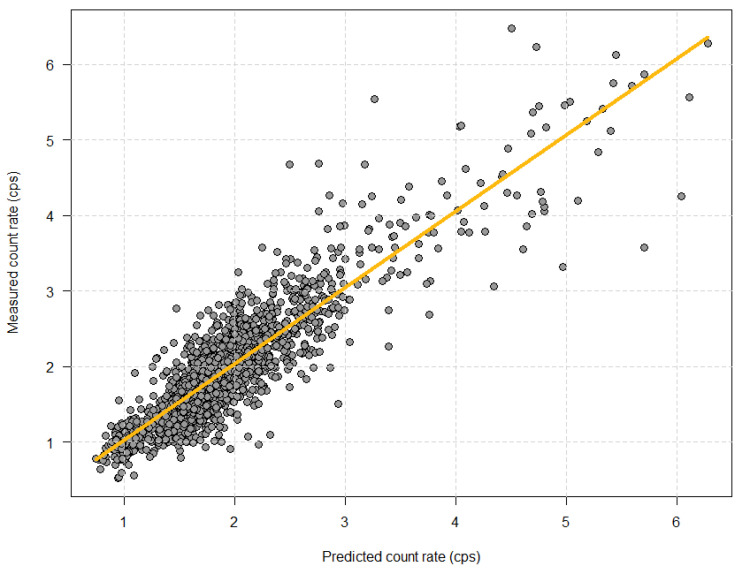
Predicted versus measured count rate.

**Table 1 diagnostics-11-02268-t001:** Detailed information on patient flow with regard to target count rate and cameras used for ERNA.

Target Count Rate (cps)	1.0	1.5	1.7	2.0	2.7	Total
Equations used	1 and 2	1 and 2	1 and 2	1 and 2	1	
Patients referred for ERNA	328	303	444	323	298	1696
Injection of FDG prior to ERNA	0	0	0	0	1	1
NaI SPECT	4	0	2	2	1	9
NaI planar	4	6	5	3	0	18 *
Dose other than scheduled	2	5	3	3	5	18 **
550 MBq due to missing data on height and weight	4	6	2	30	41	83
Received scheduled dose and CZT ERNA	314	286	432	285	250	1567
Minimum dose of 150 MBq	0	15	0	1	19	35
Minimum dose of 100 MBq	36	0	0	0	0	36

* Fourteen because of inability to raise left arm, one had to be seated upright because of dyspnoea, two were claustrophobic, and one because of sternectomy causing edge detection algorithms for SPECT to fail. ** Ten were given an extra dose of tracer due to suspicion of a partially paravenous administration of tracer due to far lower count rates than expected, five cases were due to visual suspicion of a paravenous tracer injection, one case was due to the tracer being spilt, one case was due to an obvious mismatch between information on height and weight and patient appearance, and one case was due to the patient arriving 3 h later than scheduled.

**Table 2 diagnostics-11-02268-t002:** Anthropometric data, cardiac variables, dosage information, and frequency of need for more than two analyses depending on target count rate.

	Target Count Rate Group	1.0			1.5			1.7			2.0			2.7			Standard Dose (550 MBq)			All		
	Time Period	Sep–Oct 2018			Jul–Aug 2018			Nov 2018–Jan 2019			May–Jun 2018			Mar–Apr 2018			Mar 2018–Jan 2019			Mar 2018–Jan 2019		
	*n* (Female/%) *	314	215	68.5%	286	190	66.4%	432	301	69.7%	285	187	65.6%	250	163	65.2%	83	54	65.1%	1650	1110	67.3%
Row	Variable	Mean	SD	SEE	Mean	SD	SEE	Mean	SD	SEE	Mean	SD	SEE	Mean	SD	SEE	Mean	SD	SEE	Mean	SD	SEE
1	Age (years) *	59.6	15.1	0.9	61.0	14.5	0.9	61.0	14.2	0.9	60.6	14.8	0.9	61.4	13.9	0.9	61.2	14.3	1.6	60.7	14.5	0.4
2	Height (cm) *	168.9	8.9	0.5	169.2	8.6	0.5	168.5	8.7	0.4	169.5	8.6	0.5	169.2	8.9	0.6	170.0	9.2	1.0	169.0	8.8	0.2
3	Weight (cm) *	74.8	16.0	0.9	74.7	16.8	1.0	73.5	14.9	0.7	75.7	16.6	1.0	75.1	17.2	1.1	74.3	14.9	1.6	74.6	16.1	0.4
4	Heart rate (s^−1^) *	73.2	13.5	0.8	73.3	14.0	0.8	73.9	13.1	0.8	73.4	13.1	0.8	72.9	12.4	0.8	71.9	12.9	1.4	73.3	13.4	0.3
5	Systolic blood pressure (mmHg) *	124.4	18.2	1.0	124.5	18.2	1.1	126.0	18.2	0.9	123.7	18.3	1.1	124.9	17.7	1.1	125.9	16.9	1.9	124.9	18.1	0.4
6	Diastolic blood pressure (mmHg) *	73.8	10.4	0.6	74.0	9.6	0.6	75.6	10.3	0.5	73.7	10.4	0.6	74.2	10.6	0.7	75.0	9.2	1.0	74.4	10.3	0.3
7	End diastolic volume (mL) **	88.2	25.9	1.5	90.3	25.2	1.5	88.8	26.1	1.3	92.0	25.1	1.5	93.1	25.7	1.6	96.5	28.3	3.1	90.6	25.8	0.6
8	End systolic volume (mL) **	30.9	17.4	1.0	32.0	15.9	0.9	32.5	16.8	0.8	34.2	15.5	0.9	34.9	17.8	1.1	33.9	18.9	2.1	32.8	16.8	0.4
9	LVEF (%) **	66.5	11.8	0.7	65.8	10.7	0.6	64.8	11.4	0.6	63.9	10.9	0.6	64.0	11.5	0.7	66.9	12.8	1.4	65.1	11.4	0.3
10	Target dose (MBq)	182.6	91.2	5.0	249.4	103.2	6.1	258.1	93.0	4.5	309.8	107.0	6.3	371.4	140.0	8.8	536.2	24.9	2.7	282.3	132.1	3.3
11	Dose adjusted for time of injection (MBq)	182.4	91.0	5.1	250.2	105.2	6.2	257.9	93.5	4.5	308.7	107.0	6.3	370.8	139.4	8.8	532.7	32.0	3.5	281.9	132.0	3.3
12	Predicted count rate adjusted for time of injection (s^−1^)	1.20	0.26	0.01	1.68	0.24	0.01	1.78	0.23	0.01	2.07	0.26	0.02	2.49	0.49	0.03	4.00	1.01	0.11	1.92	0.72	0.02
13	Measured count rate (s^−1^)	1.20	0.33	0.02	1.64	0.39	0.02	1.84	0.42	0.02	2.13	0.41	0.02	2.60	0.68	0.04	4.01	1.16	0.13	1.96	0.82	0.02
14	Need for more than two analyses	27	8.6%		22	7.7%		21	4.9%		8	2.8%		9	3.6%		2	2.4%		89	5.4%	

* No differences between groups; ** no difference between groups after Bonferroni correction.

## Data Availability

All the data are available in anonymised form from the corresponding author.

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
