# Peer review of "An Algorithm for Individual Dosage in Cadmium–Zinc–Telluride SPECT-Gated Radionuclide Angiography"

_diagnostics, 2021, doi:10.3390/diagnostics11122268_

Round 1

Reviewer 1 Report

I would like to know if the authors  compared the LVEF by ERNA with the LVEF by ECHO , and which was the correlation in LVEF by ERNA and ECHO  between those 4,9% pts  with disagreement  in LVEF who had  a CR of 1.7 kps.

Also how is scan time affectedfrom the reduced patient doses?

Dolomger time affect patient comfort?

In how many cases a third analyzer was needed?

Was it common for the LVEF to vary more than 2% points between the two technologists?

How target count rate was related with the accuracy of the analysis between the technologists?

Reviewer 2 Report

This is a well written manuscript. There is only one suggestion, to use receiver operating chracteristics for diagnositic accuracy. There rest are minor typo and english check.